# Effect of Vitamin D Supplementation on Skeletal Muscle Volume and Strength in Patients with Decompensated Liver Cirrhosis Undergoing Branched Chain Amino Acids Supplementation: A Prospective, Randomized, Controlled Pilot Trial

**DOI:** 10.3390/nu13061874

**Published:** 2021-05-30

**Authors:** Tomomi Okubo, Masanori Atsukawa, Akihito Tsubota, Hiroki Ono, Tadamichi Kawano, Yuji Yoshida, Taeang Arai, Korenobu Hayama, Norio Itokawa, Chisa Kondo, Keiko Kaneko, Katsuhiko Iwakiri

**Affiliations:** 1Department of Internal Medicine, Division of Gastroenterology, Nippon Medical School Chiba Hokusoh Hospital, Inzai 270-1694, Japan; ma6-0154@nms.ac.jp (T.O.); yuji-y@nms.ac.jp (Y.Y.); leaf0710@nms.ac.jp (K.H.); 2Department of Internal Medicine, Division of Gastroenterology and Hepatology, Nippon Medical School, Tokyo 113-8603, Japan; h-ono0@nms.ac.jp (H.O.); k-tadamichi@nms.ac.jp (T.K.); taeangpark@yahoo.co.jp (T.A.); itokawa@nms.ac.jp (N.I.); s8042@nms.ac.jp (C.K.); catcatkeiko1009@nms.ac.jp (K.K.); k-iwa@nms.ac.jp (K.I.); 3Core Research Facilities, The Jikei University School of Medicine, Tokyo 105-0003, Japan; atsubo@jikei.ac.jp

**Keywords:** vitamin D, decompensated liver cirrhosis, sarcopenia, skeletal muscle mass index

## Abstract

Background: Sarcopenia worsens patient prognoses in chronic liver disease. This study aimed to elucidate the effects of vitamin D supplementation on skeletal muscle volume and strength in patients with decompensated cirrhosis. Methods: Thirty-three patients were entered into the study based on the criteria and then randomly assigned to two groups: Group A (*n* = 17), the control group, and Group B (*n* = 16), those who received oral native vitamin D3 at a dose of 2000 IU once a day for 12 months. Results: SMI values in Group B were significantly increased at 12 months (7.64 × 10^−3^). The extent of changes in the SMI and grip strength in Group B were significantly greater than that in Group A at 12 months (*p* = 2.57 × 10^−3^ and 9.07 × 10^−3^). The median change rates in the SMI were +5.8% and the prevalence of sarcopenia was significantly decreased from 80.0% (12/15) to 33.3% (5/15; *p* = 2.53 × 10^−2^) in Group B. Conclusions: Vitamin D supplementation might be an effective and safe treatment option for patients with decompensated cirrhosis to increase or restore the skeletal muscle volume and strength or prevent the muscle volume and strength losses.

## 1. Introduction

Sarcopenia is characterized by generalized loss of skeletal muscle volume and strength. It is divided into two categories: “primary” sarcopenia, due to aging but not due to other causes, and “secondary” sarcopenia, due to underlying diseases, such as chronic liver, kidney and inflammatory diseases, as well as malignant tumors [1,2]. Sarcopenia, especially primary sarcopenia, causes a decline in physical fitness and health-related quality of life among community-dwelling elderly individuals [3]. Secondary sarcopenia worsens patient prognoses in various diseases, such as chronic obstructive pulmonary disease, malignant tumors and liver cirrhosis [4,5,6,7]. For patients with cirrhosis, early diagnosis and appropriate treatment for sarcopenia are important [8]. In Japan, patients with chronic liver disease are diagnosed with sarcopenia based on the criteria proposed by the Japan Society of Hepatology (JSH) [9]. The reported strategies for treating sarcopenia related to chronic liver disease include exercise regimens and supplementation with branched-chain amino acid (BCAA) [4,8,10,11,12]; however, a definitive treatment has not yet been established. For instance, exercise may induce some adverse effects, such as increased ammonia production in the muscle and elevated portal pressures [13,14]. The prevalence of sarcopenia is high in patients with advanced fibrosis [3,9,15]; thus, an effective and safe treatment for sarcopenia in patients with cirrhosis is urgently needed.

Vitamin D is a group of fat-soluble secosteroids and plays a crucial role in cell proliferation and differentiation, bone calcification, bone growth and remodeling, nerve and muscle functions, immune regulation and inflammation improvement [16]. The actions of vitamin D are mediated via the vitamin D receptor (VDR), which is a member of the nuclear receptor family of transcription factors and is expressed in systemic tissues, including the skeletal muscle [17]. We previously reported that patients with chronic liver diseases have low vitamin D levels [18,19], affecting the response to interferon therapy in patients with chronic hepatitis C [20]. Vitamin D is also involved in the development of arteriosclerosis in patients with non-alcoholic fatty liver disease [21]. Furthermore, low vitamin D is an independent factor for sarcopenia in patients with chronic liver disease [15,22]. The skeletal muscle cells strongly express the VDR [23,24], which is necessary to maintain muscle volume [25]. In the elderly, vitamin D deficiency is associated with reduced muscle volume, muscular strength and motor function, which results in an increased risk of falling [26,27,28,29,30]. Furthermore, vitamin D deficiency is an independent factor for sarcopenia in the elderly [31]. Although vitamin D supplementation improves the appendicular muscle volume and reduces the frequency of falls in elderly individuals [32,33], to the best of our knowledge, no study as of yet has investigated the influence of vitamin D supplementation on the skeletal muscle volume and strength in patients with chronic liver disease. Specifically, vitamin D deficiency and sarcopenia develop more frequently in patients with cirrhosis [9,15,34]. Thus, the improvement of these unfavorable conditions is essential for such patients.

In the present study, we aimed to elucidate the effects of vitamin D supplementation on skeletal muscle volume and strength in patients with decompensated cirrhosis, who are most frequently complicated by vitamin D deficiency and reduced skeletal muscle volume and strength.

## 2. Materials and Methods

### 2.1. Patients

Patients with decompensated cirrhosis who had been receiving oral BCAA preparations at Nippon Medical School Chiba Hokusoh Hospital (Chiba, Japan) between March 2017 and March 2019 were enrolled in the study. They were randomly assigned to two groups: Group A, the control group with no vitamin D supplementation, and Group B, those who received oral vitamin D supplements. This study was an open-label study. The leading inclusion criteria were as follows: (1) presence of decompensated cirrhosis; (2) age ≥ 20 years; (3) oral BCAA treatment that had already been initiated 6 months before the date of entry and would be continued during the study period. We routinely administered BCAA preparations to patients with decompensated cirrhosis to improve hypoalbuminemia (≤3.5 g/dL). Thus, almost all of the patients with decompensated cirrhosis received BCAA preparations at our hospital. Therefore, receiving BCAA treatment for a certain period was added to the inclusion criteria to eliminate or reduce the influence of BCAA. The exclusion criteria were as follows: (1) serum 25-hydroxyvitamin D [25(OH)D] level ≥ 30 ng/mL; (2) uncontrolled ascites; (3) uncontrolled malignant tumors including hepatocellular carcinoma (HCC); (4) jaundice or hyperbilirubinemia (≥3 mg/dL); (5) grade 3 or 4 hepatic encephalopathy; (6) chronic renal failure; (7) hypercalcemia; (8) hyperparathyroidism; (9) pacemaker use; (10) vitamin D supplementation within 6 months before the date of entry.

### 2.2. Study Protocol

Of the 102 adult patients with cirrhosis, 57 had decompensated cirrhosis to which BCAA preparations (LIVACT^®^; EA pharma, Tokyo, Japan) were administered. Based on the criteria described above, 24 patients were excluded and 33 patients were included in this study. The 33 patients were randomly assigned to Group A (*n* = 17) or Group B (*n* = 16) after enrollment, using a random number table (Figure 1). The sample size was calculated to be *n* = 18.4 with the problem probability set to 0.15 and the confidence level set to 95%, which was used as a reference for the sample size. Immediately after the allocation, the patients in Group B received oral native vitamin D3 (Now^®^ 244 Knollwood Dr, Bloomingdale, IL 60108, USA) at a dose of 2000 IU once a day for 12 months. We administered oral native vitamin D3 at a dose of D in simeplever and pegylated interferon/ribavirin combination therapy in hepatitis C patients and set the same amount this time because it was safe and effective [20]. The patients in both groups underwent physical and laboratory examinations, including measurement of serum 25(OH)D levels, skeletal muscle mass index (SMI) and grip strength, every 3 months during the study period. Patients’ adherence to vitamin D supplementation was also calculated. Day 0 was set as the time of entry in Group A and the initiation of vitamin D supplementation in Group B. Primary outcomes were changes in SMI and grip strength after 12 months and secondary outcomes were frequency of sarcopenia and safety.

### 2.3. Laboratory Methods

Hematological, biochemical and clotting tests (Table 1), which were routinely measured using conventional methods, were conducted on day 0. Serum 25(OH)D levels, a representative marker of vitamin D status, were measured using a double-antibody radioimmunoassay kit (SRL, Tokyo, Japan). Vitamin D deficiency was defined as a serum 25(OH)D level of ≤20 ng/mL [35].

### 2.4. Diagnosis of Sarcopenia

Grip strength was measured using the Smedley spring-type grip force meter. Two measurements were obtained from each hand and the average of the higher right- and left-sided values was recorded as the grip strength value. Bioelectrical impedance analysis was conducted using InBody 270 (Biospace, Seoul, Korea) to estimate the appendicular skeletal muscle volume, which was calculated as the sum of the bilateral upper and lower extremity lean muscle volume. The SMI, which is a validated indicator of sarcopenia, was calculated as follows: SMI (kg/m^2^) = appendicular skeletal muscle volume (kg)/[height (m)]^2^. The diagnosis of sarcopenia was based on the criteria proposed by the JSH [3]. According to the JSH criteria, sarcopenia is defined as the presence of decreased handgrip strength (<26 kg for men and <18 kg for women) and decreased muscle mass (SMI < 7.0 kg/m^2^ for men and <5.7 kg/m^2^ for women).

### 2.5. Ethical Statement

The present study followed the ethical guidelines established in accordance with the 2013 Declaration of Helsinki and was approved by the Institutional Review Board of Nippon Medical School Foundation (approval number: nms-2019-0102-01). All patients provided written informed consent.

### 2.6. Statistical Analyses

The Fisher exact test was used to compare frequencies in categorical data between two groups. Continuous variables with skewed distribution were compared between two groups using the Mann–Whitney test. The Wilcoxon-rank sum test was used to compare two matched samples on the same case. The level of statistical significance was set at *p* < 0.05. All statistical analyses were performed using SPSS version 17.0 software (IBM Japan, Tokyo, Japan).

## 3. Results

### 3.1. Patient Characteristics

One patient in Group B was excluded from the present analysis due to death from liver failure at 6 months. Table 1 presents the baseline characteristics of Group A (*n* = 17) and Group B (*n* = 15). Gamma-glutamyltransferase, total bilirubin and SMI significantly differed between the two groups. No other significant differences were observed.

### 3.2. Changes in Serum 25(OH)D, Albumin and Prothrombin Time

Serum 25(OH)D levels did not significantly change between day 0 and 12 months in Group A (median (range), 15.0 (5.4–25.5) ng/mL vs. 14.3 (4.7–20.6) ng/mL; *p* = 0.193). Contrarily, serum 25(OH)D levels were significantly increased in Group B (13.2 (6.1–19.2) ng/mL vs. 34.4 (18.5–43.9) ng/mL; *p* = 9.82 × 10^−4^ Figure 2). Serum 25(OH)D levels were <20 ng/mL at day 0, which corresponds to the definition of vitamin D deficiency [34], and were increased at 12 months in all the patients in Group B. Serum 25(OH)D levels were ≥20 ng/mL in all but one patient in Group B at 12 months. No significant changes in serum albumin and prothrombin time were observed in either of the groups.

### 3.3. Changes in the SMI, Grip Strength, BMI, FFM and PBF

In Group A, the SMI and grip strength did not significantly change between day 0 and 12 months (*p* = 0.158 and 0.906, respectively). In Group B, the SMI were significantly increased from day 0 to 12 months (*p* = 7.64 × 10^−3^; Figure 3). The maximum/minimum values and the first/third quartiles (but not median) of grip strength were increased from day 0 to 12 months, although the difference was not statistically significant (*p* = 0.463). In Group A, the body mass index (BMI) (median (range), 24.0 (15.0–31.6) kg/m^2^ vs. 25.3 (14.9–32.8) kg/m^2^; *p* = 0.507)), fat free mass (FFM) (median (range), 43.3 (34.4–56.0) kg vs. 42.6 (32.3–57.1) kg; *p* = 0.779)) and percentage of body fat (PBF) (median (range), 31.2 (19.3–46.5)% vs. 31.5 (17.5–46.9)%; *p* = 0.285)) did not change significantly between day 0 and 12 months. Similarly in Group B, the BMI (median (range), 22.0 (17.6–27.3) kg/m^2^ vs. 22.3 (17.3–27.1) kg/m^2^; *p* = 0.721)), FFM (median (range), 34.4 (31.0–56.4) kg vs. 35.5 (29.3–56.3) kg; *p* = 0.553)) and PBF (median (range), 33.9 (13.1–46.3)% vs. 33.8 (12.7–46.0)%; *p* = 0.678)) did not change significantly between day 0 and 12 months.

### 3.4. Comparison of the Changes in the SMI and Grip Strength between Group A and Group B

The extent of changes in the SMI from day 0 to 12 months in Group B was significantly greater than that in Group A (*p* = 2.57 × 10^−3^; Figure 4). The extent of changes in grip strength from day 0 to 12 months in Group B was significantly higher than that in Group A (*p* = 9.07 × 10^−3^; Figure 4).

### 3.5. Time Course of the Change Rates in the SMI and Grip Strength

The change rates in the SMI from day 0 were −1.8% at 6 months and −3.3% at 12 months in Group A. Contrarily, the median change rates in the SMI were +5.4% at 6 months and +5.8% at 12 months in Group B. Significant differences in the change rates between the two groups were observed at 6 months and 12 months (*p* = 2.59 × 10^−3^ and 8.57 × 10^−4^, respectively; Figure 5). Meanwhile, the median change rates in grip strength from day 0 were 0% at 6 and 12 months in Group A and 0% at 6 months and +8.0% at 12 months in Group B. A significant difference in the change rates between the two groups was observed at 12 months (*p* = 1.40 × 10^−2^; Figure 5). 

These results demonstrated that vitamin D supplementation increased the skeletal muscle volume or restored the muscle volume and increased the muscle strength in patients with decompensated cirrhosis receiving BCAA preparations.

### 3.6. Prevalence of Sarcopenia in Group A and Group B

Sarcopenia was diagnosed in 8 patients in Group A (8/17; 47.1%) and in 12 patients in Group B (12/15; 80.0%). In Group A, 3 patients were newly diagnosed with sarcopenia at 12 months, while the 8 patients with sarcopenia at day 0 remained sarcopenic at 12 months. Thus, the prevalence of sarcopenia was increased to 64.7% (11/17) at 12 months, although the difference was not significant (*p* = 0.491). Contrarily, 7 out of 12 patients diagnosed with sarcopenia at day 0 no longer met the sarcopenia diagnostic criteria at 12 months in Group B. Moreover, no patient was newly diagnosed with sarcopenia at 12 months in Group B. Thus, the prevalence of sarcopenia was significantly decreased from 80.0% (12/15) to 33.3% (5/15; *p* = 2.53 × 10^−2^) at 12 months (Figure 6). These results suggest that vitamin D supplementation may be useful for treating sarcopenia in patients with decompensated cirrhosis receiving BCAA preparations.

### 3.7. Adverse Events and Oral Compliance

No significant changes in the levels of calcium, inorganic phosphorus and creatinine were observed between day 0 and 12 months in Group A or Group B (Table 2). The patients demonstrated no symptoms due to hypercalcemia, such as anorexia, diarrhea, constipation, nausea, vomiting, sleepiness, headache, muscle pain, thirstiness, weakness and renal calculi. Patients’ adherence to vitamin D supplementation was excellent (adherence rates: 95–100%). One patient was excluded from the current analysis due to death from liver failure at 6 months, but this case eventually died of liver failure due to hepatorenal syndrome type 1 caused by cholecystitis. Therefore, it is considered that it is not related to Vitamin D administration. 

## 4. Discussion

Exercise therapy and BCAA preparations are useful for preventing and treating sarcopenia in patients with chronic liver disease [4,36,37]. In the clinical practice guideline recently established by the European Association for the Study of the Liver, the administration of vitamin D is recommended for patients with cirrhosis accompanied by vitamin D deficiency [38]. This is the first prospective, randomized, controlled pilot study to demonstrate that vitamin D supplementation significantly increased muscle volume and grip strength in patients with decompensated cirrhosis who had been receiving BCAA preparations.

The annual rates of skeletal muscle volume loss were reported to be 2.2% for patients with cirrhosis and 1.3%, 3.5% and 6.1% for those with Child–Pugh class A, B and C, respectively. These data indicate that muscle volume loss is correlated with the extent of hepatic functional reserve impairment [4]. In the present study, a decrease in the SMI after 12 months was observed in 10 (58.8%) of the 17 untreated patients (Group A) and the annual rate of muscle volume loss was 3.3%, similar to patients with Child–Pugh class B [4]. Contrarily, only 1 (6.7%) of the 15 patients supplemented with vitamin D (Group B) exhibited a decrease in the SMI after 12 months; the SMI increased at an average of 5.8% annually in this group. These results suggest that vitamin D supplementation inhibits the decrease in muscle volume or restores lost muscle volume.

The survival rate of patients with cirrhosis accompanied by sarcopenia is lower compared with that of those without sarcopenia, according to a previous report [4]. Therefore, the prevention and treatment of sarcopenia are crucial in improving the survival rate. In Japan, the prevalence of sarcopenia in patients with chronic liver disease ranges from 11% to 70% [4,15,39,40,41,42]. Our previous study also revealed that the prevalence of sarcopenia in patients with cirrhosis was 36.8% and higher than that in patients without cirrhosis [15].

In the vitamin D-supplemented group (Group B), 7 of the 12 patients who were diagnosed with sarcopenia at day 0 no longer met the diagnostic criteria for sarcopenia after 12 months due to an increase in the SMI. However, we failed to identify the factors associated with increased muscle volume (data not shown). No significant changes in grip strength were observed over time even in Group B, a finding which is consistent with previous reports [33]. However, the grip strength in Group B was significantly increased compared with that in Group A (untreated group). These results suggest that vitamin D supplementation is useful for treating or preventing sarcopenia as it increases the skeletal muscle volume/strength or restores the muscle volume/strength.

Although the mechanism by which vitamin D supplementation increases the skeletal muscle volume remains unclear, some hypotheses have been proposed for the relationship between vitamin D-related factors and sarcopenia. First, vitamin D deficiency induces atrophy of type II fibers (fast muscle fibers) in the skeletal muscle, thus resulting in sarcopenia [43]. Second, the number of VDR in the skeletal muscle decreases with age [23,44], which results in reduced vitamin D activity in the muscle. Vitamin D supplementation may increase or restore the expression of VDR [45]. A randomized controlled trial revealed that vitamin D3 administration for 4 months markedly increased the expression of nuclear VDR in elderly women with low physical performance; a strong correlation between the serum 25(OH)D and VDR expression levels was noted [46]. Third, VDR ablation and vitamin D deficiency may increase the myostatin levels [47]. Myostatin, a cytokine belonging to the transforming growth factor β1 family, inhibits the formation of skeletal muscle and, thus, is a negative regulator for muscular hypertrophy [48]. Patients with cirrhosis have higher circulating myostatin levels compared with the control subjects [49].

Adverse reactions to excessive vitamin D consumption include gastrointestinal disorders (such as anorexia, diarrhea, constipation, nausea and vomiting), sleepiness, headache, muscle pain, thirstiness, weakness and renal calculi, most of which are related to hypercalcemia [50]. Vitamin D toxicity is rare unless the serum 25(OH)D level is ≥150 ng/mL [51]. The present study and other double-blind randomized controlled trials revealed that vitamin D3 supplementation (2000 and 2800 IU per day, respectively) effectively increased the serum 25(OH)D levels even in patients with cirrhosis without inducing adverse effects [52]. Notably, we demonstrated that a relatively low dose (2000 IU per day) is safe and effective for improving vitamin D deficiency and increasing the skeletal muscle volume/strength, even in patients with decompensated cirrhosis. Patient compliance with vitamin D supplementation was found to be excellent. However, the optimal dose for treating patients with cirrhosis accompanied by muscle volume/strength loss remains unclear.

The present study has several limitations. First, this is a pilot study and includes a small number of patients. Second, the study is limited to patients with decompensated cirrhosis receiving BCAA preparations; the effects of vitamin D supplementation alone are unclear. Third, the daily physical activity and exercise in patients before and after treatment intervention were not investigated in detail. Lastly, as described above, the optimal dose of vitamin D supplementation was not examined. Given that sarcopenia in patients with chronic liver disease is associated with low vitamin D levels [15], vitamin D supplementation is rational and may be beneficial for such patients. A large-scale randomized controlled study is thus needed to confirm the effects of vitamin D supplementation on the loss of skeletal muscle volume and strength in patients with chronic liver disease, including those with decompensated cirrhosis who are treated with and without BCAA preparations.

In conclusion, the present study suggests that vitamin D supplementation might be an effective and safe treatment option for patients with decompensated cirrhosis to increase or restore the skeletal muscle volume and strength or prevent skeletal muscle volume and strength losses.

## Figures and Tables

**Figure 1 nutrients-13-01874-f001:**
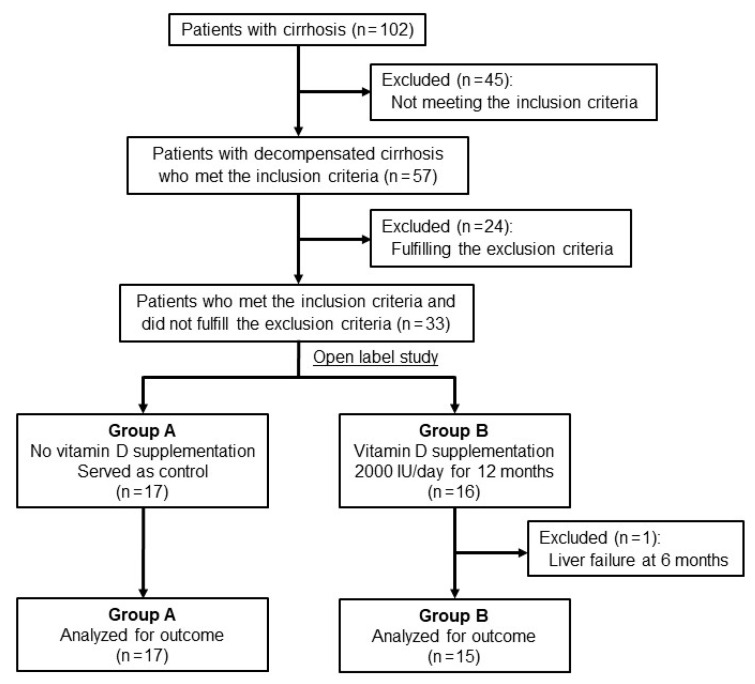
Flow diagram of patients included in the present study.

**Figure 2 nutrients-13-01874-f002:**
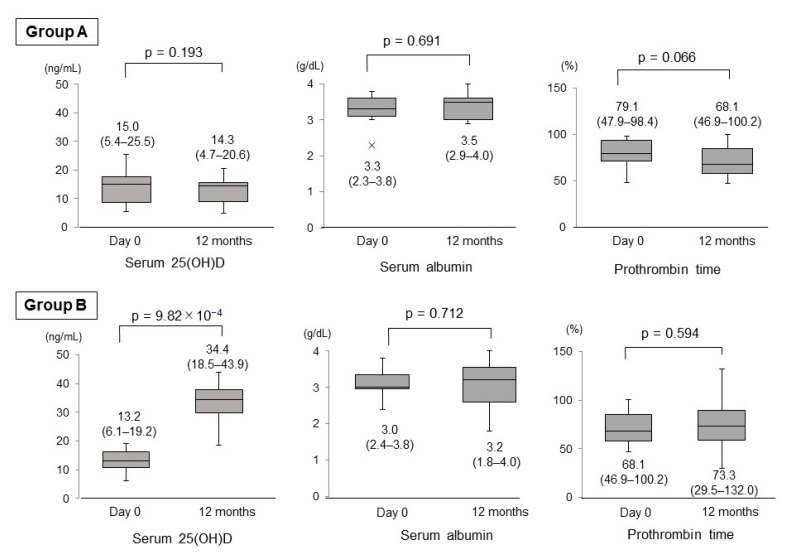
Changes in the levels of serum 25(OH)D, serum albumin and prothrombin time between day 0 and 12 months in the untreated (**Group A**) and vitamin D-treated (**Group B**) patients.

**Figure 3 nutrients-13-01874-f003:**
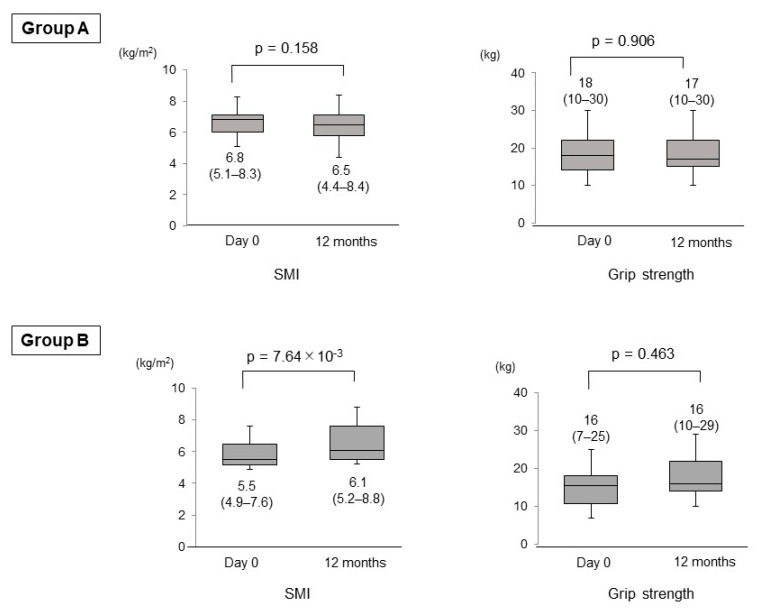
Changes in the skeletal muscle mass index and grip strength between day 0 and 12 months in the untreated (**Group A**) and vitamin D-treated (**Group B**) patients.

**Figure 4 nutrients-13-01874-f004:**
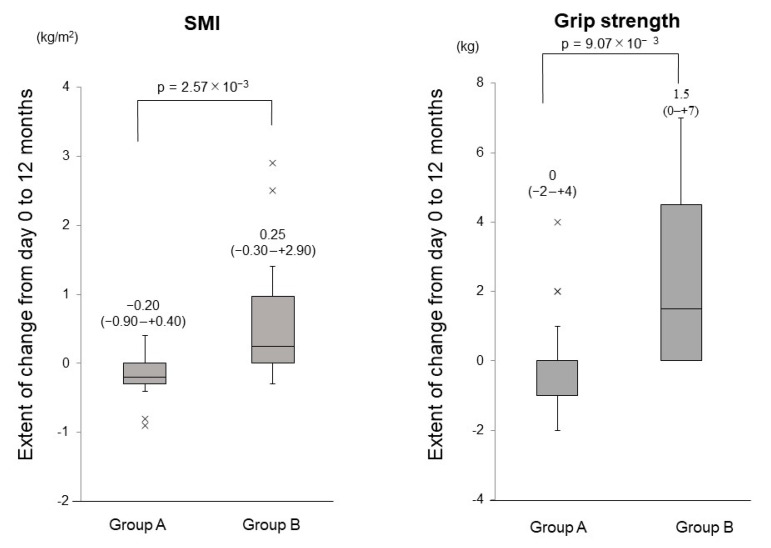
The extent of changes in the skeletal muscle mass index (SMI) and grip strength between day 0 and 12 months in the untreated (Group A) and vitamin D-treated (Group B) patients.

**Figure 5 nutrients-13-01874-f005:**
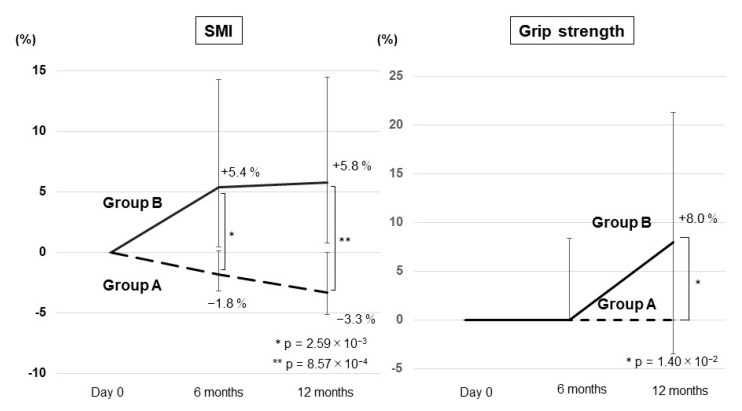
Time-course of the change rates in the skeletal muscle mass index (SMI) and grip strength in the untreated (Group A) and vitamin D-treated (Group B) patients. The error bar represents the quartile.

**Figure 6 nutrients-13-01874-f006:**
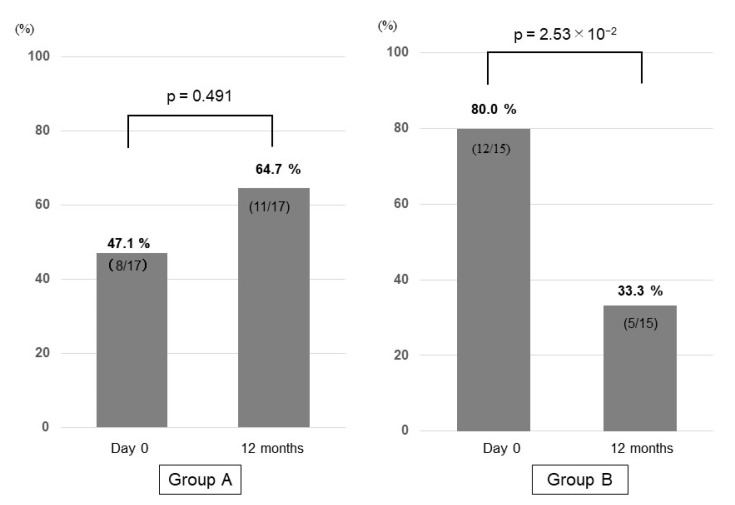
Prevalence of sarcopenia at day 0 and 12 months in the untreated (Group A) and vitamin D-treated (Group B) patients.

**Table 1 nutrients-13-01874-t001:** Baseline patient characteristics of control group (Group A) and vitamin D supplementation group (Group B).

Factors	Group A (*n* = 17)	Group B (*n* = 15)	*p* Value
Age (years)	70 (55–88)	73 (64–86)	0.226
Gender (male/female)	7/10	6/9	1.000
BMI (kg/m^2^)	24.0 (15.0–31.6)	22.0 (17.6–27.3)	0.109
Etiology of chronic hepatitis HCV/HBV/alcohol/NAFLD/PBC/AIH	7/1/6/2/1/0	8/0/2/3/1/1	-
History of HCC treatment (yes/no)	5/12	4/11	1.000
Leukocytes (/mm^3^)	4170 (2250–5980)	4700 (2980–5980)	0.428
Hemoglobin (g/dL)	13.2 (10.0–15.6)	12.2 (7.9–14.6)	0.184
Platelets (×10^3^/mm^3^)	86 (39–169)	156 (55–213)	0.079
AST (U/L)	42 (23–143)	25 (16–99)	0.104
ALT (U/L)	18 (9–43)	17 (7–63)	0.241
γ-GTP (U/L)	61 (18–374)	23 (11–72)	6.10 × 10^−4^
Calcium (mg/dL)	8.9 (7.9–10.1)	8.9 (7.9–10.0)	0.138
Inorganic phosphorus (mg/dL)	3.2 (2.4–3.6)	3.3 (2.0–3.8)	0.816
Total bilirubin (mg/dL)	1.3 (0.5–2.4)	0.8 (0.5–2.2)	2.06 × 10^−2^
Serum albumin (g/dL)	3.3 (2.3–3.8)	3.0 (2.4–3.8)	0.117
Total cholesterol (mg/dL)	153 (112–206)	181 (99–222)	0.095
Serum creatinine (mg/dL)	0.67 (0.42–2.91)	0.75 (0.52–1.41)	0.590
Prothrombin time (%)	79.1 (47.9–98.4)	68.1 (46.9–100.2)	0.443
Alpha-fetoprotein (ng/mL)	7.20 (1.02–124.90)	2.57 (1.39–27.24)	0.279
WFA^+^-M2BP (C.O.I)	3.96 (1.53–12.35)	2.88 (1.07–13.01)	0.702
Serum 25(OH)D3 (ng/mL)	15.0 (5.4–25.5)	13.2 (6.1–19.2)	0.606
BCAA administration period (months)	45 (4–80)	23 (9–46)	0.089
Grip strength (kg)	18 (10–30)	16 (7–25)	0.157
SMI (kg/m^2^)	6.8 (5.1–8.3)	5.5 (4.9–7.6)	1.10 × 10^−2^
FFM (kg)	43.3 (34.4–56.0)	34.4 (31.0–56.4)	0.212
PBF (%)	31.2 (19.3–46.5)	33.9 (13.1–46.3)	0.935
Sarcopenia (yes/no)	8/9	12/3	0.120

Categorical variables are given as numbers. Continuous variables are given as medians and ranges in parentheses. BMI, body mass index; HCV, hepatitis C virus; HBV, hepatitis B virus; NAFLD, nonalcoholic fatty liver disease; PBC, primary biliary cholangitis; AIH, autoimmune hepatitis; HCC, hepatocellular carcinoma; AST, aspartate aminotransferase; ALT, alanine aminotransferase; γ-GTP, gamma-glutamyltransferase; WFA^+^-M2BP, Wisteria floribunda agglutinin positive Mac-2-binding protein; COI, cut-off index; 25(OH)D3, 25-hydroxyvitamin D3; BCAA, branched-chain amino acid; SMI, skeletal muscle mass index; FFM, fat free mass; PBF, percentage of body fat.

**Table 2 nutrients-13-01874-t002:** Comparison between day 0 and 12 months.

Comparison between day 0 and 12 months in control group (Group A)			
**Factors**	**Day 0**	**12 Months**	***p* Value**
Calcium (mg/dL)	8.9 (7.9–10.1)	8.9 (7.9–10.0)	0.650
Inorganic phosphorus (mg/dL)	3.2 (2.4–3.6)	3.3 (2.0–3.8)	0.500
Serum creatinine (mg/dL)	0.67 (0.42–2.91)	0.77 (0.46–1.86)	0.064
**Comparison between day 0 and 12 months in vitamin D supplementation group (Group B)**			
**Factors**	**Day 0**	**12 Months**	***p* Value**
Calcium (mg/dL)	8.9 (7.9–10.0)	9.0 (7.7–10.0)	0.109
Inorganic phosphorus (mg/dL)	3.3 (2.0–3.8)	3.2 (2.9–3.7)	0.317
Serum creatinine (mg/dL)	0.75 (0.52–1.41)	0.95 (0.54–2.63)	0.173

## Data Availability

The data presented in this study are available upon request from the corresponding author. The data are not publicly available due to privacy and ethical reasons.

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
