# Peer review of "Effect of Vitamin D Supplementation on Skeletal Muscle Volume and Strength in Patients with Decompensated Liver Cirrhosis Undergoing Branched Chain Amino Acids Supplementation: A Prospective, Randomized, Controlled Pilot Trial"

_nutrients, 2021, doi:10.3390/nu13061874_

Round 1

Reviewer 1 Report

an interesting article about the use of vitamin D supplementation in patients with decompensated cirrhosis receiving branched chain amino acids.  The main problem of the study is the low number of participants, as 33 patients are not enough to reach a good grade of statistical significance, making a bigger trial necessary in order to confirm these findings; I have some queries:

The title should be modified reporting BCAA supplementation: "Effect of vitamin D supplementation on skeletal muscle volume and strength in patients with decompensated liver cirrhosis undergoing branched chain amino acids supplementation: A prospective, randomized, controlled pilot trial

page 2 line 47-49 "Vitamin D is a group of fat-soluble secosteroids and plays a crucial role in cell proliferation and differentiation, bone calcification, bone growth and remodeling, nerve and muscle functions, immune regulation, and inflammation improvement." this paragraph needs a reference, such as: doi: 10.1007/s13668-020-00322-4.

Thank You

Author Response

We were pleased to read the fair assessment of the reviewer and the encouraging comments by expert reviewers. We believe that adequate modifications were successfully made in our revision, taking into consideration the comments raised by the expert reviewer. We also carefully proofread the revision including the text, figure, table and references.

Reviewer 1.

  1. As suggested by the reviewer, we changed the title.
  2. As suggested by the reviewer, we added reference to the paragraphs pointed out in page 2.

Reviewer 2 Report

Dear Authors,

thank you for allowing me to read this interesting paper.

I do not see any attached figures nor tables so I can not comment on them - please provide them in the main body.

The introductions, importance and overall goal are well described.

I would like you to add information if it was a blinded - single or double and did the A group receive any placebo? 

It would be good to present a flowchart from the whole intervention.

In my opinion it would be worth to show changes in anthropometric parameters (BMI, % fat mass?, % ffm?) to see whether they were impacted by vit D supplementation as was the SMI and grip strength.

As the study group is not big it was not (as I understand) a blinded study I would reconsider softening final conclusions

Aside from from that, it is an interesting a worthy study

Author Response

We were pleased to read the fair assessment of the reviewer and the encouraging comments by expert reviewers. We believe that adequate modifications were successfully made in our revision, taking into consideration the comments raised by the expert reviewer. We also carefully proofread the revision including the text, figure, table and references.

  1. As suggested by the reviewer, this study was not blind test. We added the following sentence in page 2 line 79 and added to the flowchart of Figure 1.

“This study was an open-label study.”

  1. As suggested by the reviewer, the changes over time in BMI, FFM, and PBF were added to the result section in page 4 line 157. In addition, the baseline FFM and PBF have been added to Table 1.
  2. As suggested by the reviewer, we changed the sentence of conclusion in abstract (page 1 line 25) and text (page 6 line 282) section.